# Functional Role of miR-155 in the Pathogenesis of Diabetes Mellitus and Its Complications

**DOI:** 10.3390/ncrna7030039

**Published:** 2021-07-07

**Authors:** Stanislovas S. Jankauskas, Jessica Gambardella, Celestino Sardu, Angela Lombardi, Gaetano Santulli

**Affiliations:** 1Department of Medicine, Fleischer Institute for Diabetes and Metabolism (FIDAM), Einstein-Mount Sinai Diabetes Research Center (ES-DRC), Albert Einstein College of Medicine, New York, NY 10461, USA; stanislovas.jankauskas@einsteinmed.org (S.S.J.); jessica.gambardella@einsteinmed.org (J.G.); angela.lombardi@einsteinmed.org (A.L.); 2Department of Molecular Pharmacology, Wilf Family Cardiovascular Research Institute, Einstein Institute for Aging Research, Albert Einstein College of Medicine, New York, NY 10461, USA; 3International Translational Research and Medical Education Consortium (ITME), Department of Advanced Biomedical Science, “Federico II” University, 80131 Naples, Italy; 4Department of Advanced Medical and Surgical Sciences, University of Campania “Luigi Vanvitelli”, 80138 Naples, Italy; celestino.sardu@unicampania.it

**Keywords:** β-cells, diabetes, epigenetics, inflammation, insulin, islets, MafB, metabolic syndrome, micro-RNA, miR-155, NF-κB, NRF2, PBMC

## Abstract

Substantial evidence indicates that microRNA-155 (miR-155) plays a crucial role in the pathogenesis of diabetes mellitus (DM) and its complications. A number of clinical studies reported low serum levels of miR-155 in patients with type 2 diabetes (T2D). Preclinical studies revealed that miR-155 partakes in the phenotypic switch of cells within the islets of Langerhans under metabolic stress. Moreover, miR-155 was shown to regulate insulin sensitivity in liver, adipose tissue, and skeletal muscle. Dysregulation of miR-155 expression was also shown to predict the development of nephropathy, neuropathy, and retinopathy in DM. Here, we systematically describe the reports investigating the role of miR-155 in DM and its complications. We also discuss the recent results from *in vivo* and *in vitro* models of type 1 diabetes (T1D) and T2D, discussing the differences between clinical and preclinical studies and shedding light on the molecular pathways mediated by miR-155 in different tissues affected by DM.

## 1. Introduction

The pandemic of diabetes mellitus (DM) represents one of the most important long-term challenges for healthcare systems around the globe [1]. The International Diabetes Federation (IDF) has estimated that DM will affect ~642 million people by 2040 [2]. DM is accountable for 6.8% of global mortality in an age group of 20–79 years old and constitutes a major cause of reduced life expectancy [3,4,5]. Approximately 10% of DM patients suffer from type 1 diabetes (T1D), caused by an autoimmune destruction of pancreatic β-cells [6,7], while 90% of DM cases refer instead to type 2 diabetes (T2D), whose main risk factors are obesity, lack of physical exercise, and smoking [1,8]. Several clinical trials have demonstrated that lifestyle interventions effectively prevent T2D development [9,10,11,12]. However, hereditary factors have also been identified as important modulators of individual susceptibility for T2D development and response to lifestyle interventions [13,14,15]. These facts support the demand for further research on the molecular bases of DM and elucidating new factors involved in DM pathogenesis.

The past decades brought high attention to microRNAs (miRNAs) as new fine regulators of gene expression, miRNAs are small (~22-nucleotide long) non-coding one-stranded RNAs harboring a “seed” sequence at 5′-end, which is usually 6–8 nucleotides long and binds the 3′ untranslated region (3′UTR) of specific target messenger RNA (mRNA). Upon binding to the mRNA 3′UTR, miRNAs either block translation from this mRNA or facilitate its degradation, thereby repressing gene expression [16,17,18,19].

Thousands of mammalian mRNAs have been predicted to be targeted by miRNAs and hundreds of such interactions have been experimentally proven so far [20,21]. An overwhelming amount of data demonstrated that miRNAs fine-tuning of gene expression plays indispensable roles in human development and disease [22,23,24,25,26,27,28,29].

## 2. miR-155 Biogenesis

First discovered in chicken in 1997, miR-155 was identified as a product of a gene that was transcriptionally activated by promoter insertion at a common retroviral integration site in B-cell lymphomas and was formerly called B-cell Integration Cluster (*BIC*) [30]. Its homologs were later identified in humans and mice [31,32]. The maturation of miR-155 follows the conventional process of miRNA biogenesis (summarized in Figure 1), which starts with pri-miRNA transcription and is followed by its cleavage by the nuclear microprocessor complex (including Drosha and DGCR8 proteins) with a production of a 65-nucleotide stem-loop precursor miRNA (pre-miR-155). After being exported from the nucleus by exportin-5, pre-miR-155 molecules are cleaved again by Dicer, resulting in RNA duplexes of ~22 nucleotides. Argonaute protein binds to the short RNA duplexes, forming the core of a multi-subunit complex called RNA-induced silencing complex (RISC), generating one-stranded RNA molecules capable of mRNA binding [33,34].

Both arms of the pre-miRNA hairpin can give rise to mature miRNAs possessing the biological activity [35,36,37]. They are denoted with the suffix -5p (from the 5′ arm) (e.g., miR-155-5p) and -3p (from the 3′ arm) (e.g., miR-155-3p) [38].

## 3. Dysregulation of miR-155 in DM: Findings from Clinical Studies

The main clinical studies investigating miR-155 and DM are reported in Table 1. A significant decrease in serum levels of miR-155 in T2D patients has been shown in Italian, Chinese, Egyptian, Brazilian, and Iranian cohorts, exposing a striking consistence among the different ethnic backgrounds [39,40,41,42,43].

A reduced expression of miR-155 was observed in peripheral blood mononuclear cells (PBMCs) isolated from T2D patients compared to healthy volunteers [48,49]. Similarly, a decreased miR-155 expression was reported in platelets of diabetic patients [55]. Low plasma levels of miR-155 were also found in gestational DM [54]. Interestingly, patients with hepatitis C and insulin resistance also displayed lower serum levels of miR-155 compared to those with hepatitis C without insulin resistance, although hepatitis C infection was associated with manifold upregulation of this miRNA [41]. Plasma levels of miR-155-5p were shown to correlate with diabetic neuropathy: patients with diabetic distal symmetric sensorimotor polyneuropathy and diabetic cardiovascular autonomic neuropathy had lower plasma levels of miR-155 compared to DM patients without neurological symptoms [39]. Corral-Fernández et al. showed that the downregulation of miR-155 in PBMCs was more pronounced in patients with exaggerated obesity [48].

The expression of IL-6 and TNFα in response to LPS was found to be augmented in PBMCs isolated from T2D patients compared to cells obtained from healthy controls [49], providing a possible mechanistic link to higher plasma levels of IL-6 in T2D [48]. Of note, miR-155 expression was also found to be altered in the adipose tissue [58].

Profiling miRNAs expression in omental and subcutaneous adipose tissue revealed that miR-155 expression had a negative correlation with the mean adipocyte volume in omental fat and with macrophage infiltration in subcutaneous adipose tissue [57]. Obese non-diabetic patients displayed increased levels of miR-155 both in bloodstream and in adipose tissue, suggesting that miR-155 upregulation may play a beneficial role in maintaining adipose tissue health [58].

Surprisingly, despite the abundant data showing downregulation of miR-155 in T2D, patients suffering from T1D displayed an increased expression of miR-155 in PBMCs compared to healthy subjects [51,52]. T1D patients also had a higher prevalence of the T/T allele for the rs767649 polymorphism of miR-155, which is likely associated with a higher transcriptional activity of miR-155 [50]. Assamann and collaborators have demonstrated that miR-155-5p serum levels are robustly upregulated in freshly diagnosed T1D patients, whereas more than 5 years of disease are associated with its downregulation [45].

In contrast to other tissues, renal expression of miR-155 seems to be augmented in DM. An elegant study by Baker et al. utilized laser microdissection to evaluate the changes in miRNAs expression in the kidney, showing that diabetic nephropathy is associated with increased miR-155 expression in proximal tubules, but not in glomeruli [60]. A study made on a smaller number of patients revealed a higher expression of miR-155-5p both in renal tubules and glomeruli in patients with diabetic nephropathy compared to healthy individuals [59]. High serum levels of miR-155 were found in patients with diabetic nephropathy in two trials [46,47]. Moreover, when combined with high urinary levels of vitamin D binding protein, elevated serum miR-155 appeared to be a strong predictor of rapid disease progression [46]; these findings were accompanied by the observation of a robust increase in urinary levels of miR-155 [61].

Nevertheless, a study measuring miR-155 in urinary exosomes demonstrated that patients with DM had lower amounts of miR-155 than healthy volunteers [62]. Similarly, miR-155 levels in urinary exosomes were lower in diabetic patients with microalbuminuria than in diabetic patients with unaltered renal function [62]. Difficulties in precisely trace the origin of urinary exosomes, which could be shed from epithelial cells all along the nephron and urinary tract, may explain the contradiction between this study and other results on renal expression of miR-155.

Controversial results were reported when exploring the correlation between miR-155 levels and diabetic retinopathy. On the one hand, a clinical study revealed decreased levels of miR-155 in patients with diabetic retinopathy compared to diabetic patients with no eye abnormalities [56]. Another study identified the A-allele of rs767649 polymorphism in miR-155 gene as a strong predictor of diabetic retinopathy [43]. The A-allele of rs767649 polymorphism is associated with lower transcriptional activity of miR-155. However, plasma levels of miR-155 demonstrated no predictive power in this study [43,50].

On the other hand, a study in Chinese patients revealed a positive correlation between the severity of diabetic retinopathy and plasma levels of miR-155 [53], and a large cohort of Brazilian patients showed no association between high plasma levels of miR-155-3p and diabetic retinopathy [44].

## 4. Different Roles of miR-155 in the Pathogenesis of T1D and T2D

### 4.1. miR-155 and T2D

Pancreatic β-cell dysfunction and insulin resistance are two hallmarks of T2D [8,63,64,65,66,67,68]. Epidemiologic studies suggest that more than 60% of T2D can be attributed to obesity [69]. However, the effects of high body adiposity on β-cell function remain not fully understood [70]. The examination of human pancreatic tissue received from patients with a different disease status revealed a bell-shaped curve relationship between β-cell mass and disease progression.

At least in the initial phases of the disease, obese non-diabetic patients exhibit a 25–50% increase in β-cell mass and indirect evidence suggests an increased capacity of β-cells to release insulin [71,72,73,74,75,76,77], most likely as adaption to increased glucose uptake [77,78,79,80,81,82,83]. However, after the onset of frank hyperglycemia, β-cell mass starts to fall down, eventually reaching levels even lower than those of healthy lean individuals [79,81,82,84,85]. The insulin secretion capacity of β-cells from patients with overt T2D is also impaired, thereby further compromising insulin levels [86,87,88,89,90,91].

Preclinical studies suggest that the decreased expression of miR-155 observed in T2D patients may represent one of the mechanisms underlying the declined pancreatic function. Indeed, miR-155 whole-body knockout or overexpression in mice did not significantly affect islet number and morphology, insulin expression, or β-cell proliferation [40]. However, miR-155 deficiency markedly decreased β-cell expansion in response to hypercholesteremia and was also accompanied by diminished insulin secretion and augmented expression of glucagon, alongside a relative increase of glucagon positive cells in pancreatic islets, thus recapitulating the pathological findings of human T2D [92]. The same group of researchers found that miR-155 expression was directly upregulated in β-cells by both native and myeloperoxidase-modified low-density lipoproteins [92]. The main target of miR-155 appeared to be V-maf musculoaponeurotic fibrosarcoma oncogene homolog B (*MafB*), an established master regulator of the α-cell phenotype [92].

Another preclinical study demonstrated that upon high-fat diet a phenotype switch occurs in the islets of Langerhans, with insulin-producing β-cells re-differentiating into glucagon-producing α-cells [93]. Thus, miR-155 is a pivotal factor maintaining β-cell fitness; its upregulation represents a β-cell adaptation to overnutrition and disruption of this mechanism may be a key event in the transition from prediabetes to T2D.

A fundamental feature of T2D is insulin resistance, a condition of blunted response to insulin of tissues, mainly skeletal muscles, fat, and liver [8,94]. Skeletal muscles and adipose tissue respond to insulin by enhancing glucose uptake and thus playing a central role in the regulation of glucose homeostasis.

The liver is the main body site of endogenous glucose synthesis [95,96] and this process is inhibited by insulin [97,98]. Mice lacking miR-155 fed normal chow exhibit impaired glucose tolerance and decreased insulin sensitivity with unaltered insulin production capacity. Instead, global miR-155 overexpression ameliorates both glucose tolerance and insulin sensitivity [40]. This effect is mainly achieved through means of the downregulation of genes involved in negative modulation of insulin receptor signaling and upregulation of glucose uptake genes in adipose tissue and skeletal muscle. In the liver, miR-155 downregulates both negative modulators of insulin receptor signaling and genes involved in gluconeogenesis [40]. Indeed, miR-155 directly targets the 3′UTRs of CCAAT-enhancer-binding proteinβ (C/EBPβ) and Histone deacetylase 4 (HDAC4).

Thus, a decreased expression of miR-155 in T2D patients may be one of the fundamental factors underlying the development of insulin resistance (Figure 2).

### 4.2. miR-155 and T1D

In contrast to T2D, the primary mechanism of T1D is an autoimmune attack on β-cells, and the progressive loss of insulin producing cells underlies the disease progression. Both human and animal data have shown that the functional contribution of miR-155 in the pathogenesis of T1D could be totally different from what was observed in T2D. Results obtained in a murine model of T1D—non-obese diabetic (NOD) mice—appeared to be consistent with findings in clinical studies and demonstrated an upregulation of miR-155 [99]. This animal model also recapitulated the increased expression of miR-155 in some types of immune cells [51,52,99].

Moreover, miR-155 was markedly enriched in exosomes derived from T-lymphocytes of NOD mice. β-cells treated with these exosomes underwent apoptosis and such effect was inhibited by inhibiting miR-155. Similarly, the ectopic expression of a specific long non-coding RNA (LcnRNA) sponging miR-155 mitigated the development of DM in NOD mice [99]. These data strongly suggest that the upregulation of miR-155 in immune cells contributes to β-cell death in T1D.

Activating mutations in the Signal transducer and activator of transcription 3 (*STAT3*) have been proposed to cause T1D, most likely via impaired development of regulatory T cells (Tregs) and expansion and activation of T helper type 17 cells [100,101,102,103,104]. Interestingly, STAT3 is able to bind and upregulate the expression of the miR-155 promoter [105].

Furthermore, STAT3 dependent upregulation of miR-155 has been shown to play a crucial role in the development of several autoimmune pathologies, giving another clue to the potential deleterious role of miR-155 in the pathogenesis of T1D [106,107,108].

Another protein involved in the pathophysiology of T1D is Forkhead box protein P3 (FOXP3). Mutations in its gene are known to cause T1D [109], most likely since the FOXP3 activity is required for Tregs maturation, which in turn suppresses the cytolytic activity of CD4^+^ and CD8^+^ T cell [110]. Mice specifically overexpressing miR-155 in FOX3P3^+^ Tregs exhibit spontaneous autoimmunity [111], whereas miR-155 deficiency results in the disproportional loss of Tregs and inability to maintain immune homeostasis [112,113,114].

## 5. Physiological Roles of miR-155

Several studies have shown that miR-155 plays important roles in hematopoiesis. It was identified among five other miRNAs (i.e., miR-142, miR-144, miR-150, miR-155, and miR-223) that were specific for hematopoietic cells both in humans and in mice [35].

Further experiments demonstrated that miR-155 is highly expressed in hematopoietic stem-progenitor cells (HSPCs) and maintains cells in an undifferentiated state via direct repression of transcription factor PU.1 [115,116,117]. Forced overexpression of miR-155 markedly reduced both myeloid and erythroid colony formation of normal human HSPCs [115].

miR-155 may also play independent roles in the regulation of erythropoiesis, since its expression is drastically decreased upon differentiation of human erythroid progenitors into erythroblasts [118].

Additionally, miR-155 plays vital actions in immune cells, being essential for the differentiation of Tregs by repressing the expression of Suppressor of cytokine signaling 1 (*SOCS1*) [112,119]. miR-155 is also involved in the selection of competent B cells in germinal centers, since its deficiency has been shown to allow immature B cells to avoid apoptosis [116,120,121], and in innate immune response, playing a pro-inflammatory role: Its expression is dramatically upregulated in macrophages, dendritic cells, and other immune cells upon Toll-like receptors (TLRs) activation by pathogen associated molecular patterns [122,123,124,125,126].

Nuclear factor kappa-light-chain-enhancer of activated B cells (NF-κB) and Activator protein 1 (AP-1) were identified as two main transcription factors responsible for miR-155 transcriptional activity [122,123,127,128]; miR-155-5p suppresses negative regulators of inflammation, such as inositol polyphosphate-5-phosphatase (INPP5D, also known as SHIP1) and SOCS1, thus promoting immune cell survival, growth, migration, and anti-pathogen responses [128,129,130].

Nonetheless, some data indicate that miR-155-5p may also limit inflammatory response, suggesting that the role of miR-155 could change according to the different phases of inflammation [131]. Taking in consideration the profound effects of miR-155 on the immune system, it is not surprising that miR-155 was found to be an important factor determining the development and course of cancers, infectious diseases, and autoimmune disorders (including T1D) [132,133,134,135,136,137,138].

## 6. Functional Role of miR-155 in DM Complications

Several studies have demonstrated that miR-155 participates in the pathogenesis of diabetic complications, which include nephropathy, neuropathy, cardiomyopathy, and retinopathy [139,140,141,142,143,144,145].

### 6.1. miR-155 and Pathogenesis of Diabetic Nephropathy

A primary mechanism of renal damage in DM is uncontrolled hyperglycemia, which triggers glomerular endothelium activation and podocyte dysfunction [146,147]. The inflammatory response triggers activation of mesangial cells in the glomeruli, which start to proliferate and produce excessive extracellular matrix substituting capillaries and podocytes, thus resulting in a decline in the glomerular filtration rate. Eventually, glomerulosclerosis triggers tubular epithelium damage with irreversible fibrotic changes [145,148,149,150].

The expression of miR-155-5p was found to be increased by hyperglycemia both *in vitro* and *in vivo*, thus phenocopying the human findings of elevated renal miR-155 expression in diabetic nephropathy [47,151,152,153]. In animal studies, miR-155-5p overexpression was associated with higher interstitial fibrosis—hallmark of irreversible kidney damage—and attenuated by miR-155-5p inhibition [152,154,155].

Several signaling pathways have been implicated in the beneficial effect of miR-155 inhibition. The first one is the direct binding to the 3′UTRs of *Socs1* and *Socs6* [154], which are known to inhibit the JAK-STAT pathway, which is also involved in signaling mediated by the master regulator of renal fibrosis, transforming growth factor β1 (TGF-β1) [156,157]. Another confirmed target of miR-155-5p is NAD-dependent deacetylase sirtuin-1 (*Sirt1*). Downregulation of *Sirt1* expression by miR-155-5p is followed by suppressed autophagy, as SIRT1 dependent deacetylation is required for the activation of a number of proteins involved in autophagy signaling and execution [151,155,158,159]. The third pathway is given by the direct inhibition of *Pten*, a negative regulator of PI3K/AKT/mTOR pathway, which in turn inhibits autophagy [152]. Consistent with these findings, a number of investigations demonstrated that an impaired autophagic response undermines the kidney ability to cope with pathological stresses [160].

miR-155 may also play a role in the early stages of diabetic nephropathy. Several studies have shown that miR-155 expression is dramatically upregulated in both human and rat glomerular mesangial cells (GMCs) exposed to high-glucose [47,153]. Interestingly, miR-155 upregulation is blunted in TLR4-deficient cells, suggesting a primary role of inflammation in miR-155 transcriptional control [153]. This view is supported by the augmented miR-155 expression observed in human GMCs in response to pro-inflammatory cytokines [161]. Moreover, treatment with LncRNA CTBP1-AS2 sponging miR-155 protects GMCs by diminishing miR-155-mediated repression of *Foxo1* [47]; miR-155 inhibition was also able to reduce TGF-β1-induced signaling in podocytes *in vitro*, and a decreased expression of markers of podocyte injury was observed under miR-155 inhibition *in vivo* [162,163].

### 6.2. Role of miR-155 in the Pathogenesis of Diabetic Neuropathy and Diabetic Cardiomyopathy

Preclinical studies investigating the potential role that miR-155 could play in diabetic neuropathy have emphasized the differences between T1D and T2D. Downregulation of miR-155 was found in the sciatic nerve of *db/db* mice, an established model of T2D (119). The treatment with miR-155 mimic had profound beneficial effects characterized by augmented velocities of conduction in both sensory and motor nerves and significant decrease in the threshold to thermal stimuli; miR-155 also reversed the DM-triggered production of inflammatory cytokines and the upregulation of neurogenic locus notch homolog protein 2 (*Notch2*) (119).

Instead, a study utilizing a model of T1D revealed that miR-155 inhibition protected animals from nerve demyelination. This effect was attributed to the direct downregulation of nuclear factor erythroid 2-related factor 2 (*Nrf2*) expression by miR-155 and subsequent apoptosis of Schwann cells exposed to high-glucose (120).

*in vitro* assays have demonstrated that hyperglycemia upregulates miR-155 expression in cardiomyoblasts, leading to an increased expression of pro-fibrotic genes and development of mitochondrial dysfunction [164]. Of note, a tight glycemic control tested *in vivo* in a model of T1D failed to prevent miR-155 upregulation in the heart [165].

### 6.3. miR-155 and Diabetic Retinopathy

Diabetic eye disease affects approximately 80% of patients who have been affected by T1D or T2D for at least 20 years [166]. The disease starts with non-proliferative lesions such as altered retinal blood flow and vascular permeability, basement membrane thickening, loss of pericytes, and the formation of acellular capillaries, causing macular edema. Then, the disease enters into a proliferative stage, associated with pathological neovascularization: vessels grow into the vitreous eliciting hemorrhagic events or retinal detachment which may result into blindness [167,168].

An upregulation of miR-155 was also observed in several animal models of retinal injury [169,170]. The global ablation of miR-155 in mice did not produce significant changes to retinal architecture or retinal angiogenesis [169,170]. Nonetheless, miR-155 expression was upregulated in human retinal microvascular endothelial cells (HRMECs) upon treatment with vascular endothelial growth factor (VEGF) *in vitro* and inhibition of miR-155 abrogated VEGF-mediated proliferation, migration, and ability to form network-like structures of HRMECs [171].

Both the knockout of miR-155 and its inhibition were shown to prevent the pathological neovascularization of the retina and to attenuate vision loss [169,170,171]. Additionally, miR-155 affected endothelial function via downregulation of *Ship1*, a negative regulator of PI3K/AKT pathway, and Cysteine-rich angiogenic inducer 61 (*Cyr61*, also known as *Ccn1*), an extracellular matrix-associated integrin-binding protein that promotes angiogenesis [170,171,172,173,174].

Upregulation of miR-155 was also reported in human retinal pigment epithelium treated with pro-inflammatory cytokines [175]. Another indication of the deleterious role of high miR-155 in retinopathies came from a study showing that saponin-mediated protection of photoreceptors involves precluding miR-155 overexpression in macrophages [176].

## 7. Conclusions and Perspectives

Taken together, the data discussed above provide a strong evidence in support of miR-155 as a new key player in the pathogenesis of DM. Numerous clinical studies performed in different centers show that miR-155 downregulation is a common finding in T2D patients. Of course, large multicenter trials are required to determine if circulating levels of miR-155 could be established as a new and reliable biomarker of T2D. Considering that animal studies revealed miR-155 downregulation as a potential trigger of insulin resistance and β-cells loss, miR-155 may possibly be used for early diagnostics of T2D or prediction of T2D outcomes. 

miR-155 may also represent a new drug target. The success of COVID-19 vaccines BNT162b2 and mRNA-1273, based on mRNA delivery, could also open the possibility to use miRNA-based approaches as a medication to tackle DM [177,178,179]. Another important direction in miR-155 research is to determine the mechanisms underlying its downregulation in T2D, which should likely provide other more conventional targets for drug development.

The net effect of miR-155 downregulation or overexpression could be dissimilar in different organs. However, the experimental data from human studies provides us with a clear pattern showing that miR-155 tends to be decreased in patients with T2D and increased in T1D. Such a discrepancy could be related to the different pathogenesis of these two diseases. The findings in animal models are consistent with clinical trials and may provide the basis for potential mechanistic explanations. We speculate that miR-155 downregulation contributes to the development of insulin resistance in adipose tissue and skeletal muscle and compromises pancreatic insulin secretion by dedifferentiating β-cells. Furthermore, mounting evidence indicates that high levels of miR-155 favor progression of a number of autoimmune pathologies, including insulitis.

A relatively small amount of data is available on the role of miR-155 in T1D. Studies performed hitherto strongly suggest that miR-155 plays detrimental roles in autoimmune demise of β-cells. Nevertheless, more studies are required to establish an unambiguous association linking miR-155 and T1D. If confirmed, miR-155 may become a screening tool for detecting insulitis before frank clinical manifestations. Moreover, if miR-155 is in fact involved in autoimmune destruction of β-cells, it might be harnessed as a target for T1D treatment. Of note, drugs based on miR-155 inhibition are already under extensive development [180].

Inhibition of miR-155 may also become a therapeutic option for DM complications. Human and animal data consistently point to the essential role of miR-155 upregulation in diabetic nephropathy. Promising results have been obtained in preclinical models of diabetic neuropathy giving hope for mitigation of neuropathic pain and peripheral demyelination. Experimental models of diabetic retinopathy also hint that miR-155 inhibition may save vision for millions of DM patients. However, clinical data on diabetic retinopathy remain controversial and require further investigation. 

## Figures and Tables

**Figure 1 ncrna-07-00039-f001:**
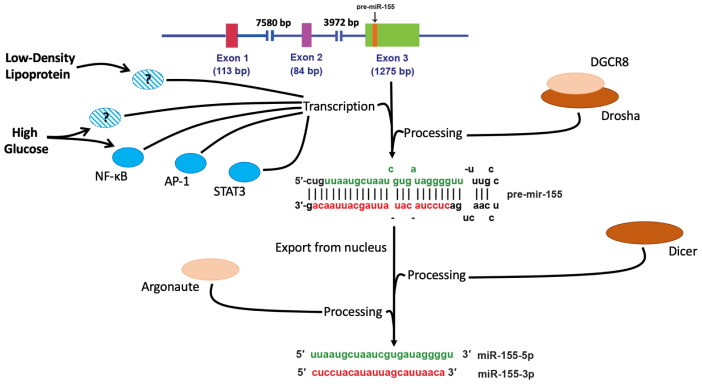
**Schematic overview of miR-155 biogenesis.** Several transcription factors (including NF-κB, AP-1, and STAT3) have been identified as positive regulators of miR-155. High-glucose and LDL also upregulate miR-155 transcription, however, the exact signaling pathways are not fully understood. The transcript is cleaved by the nuclear microprocessor complex (including Drosha and DGCR8 proteins) with a production of a 65-nucleotide stem-loop precursor miRNA (pre-miR-155); pre-miR-155 is exported from the nucleus by exportin-5 and cleaved by Dicer resulting in RNA duplexes of ~22 nucleotides. Argonaute binds to the miR-155 duplexes, forming the core of RISC complex, producing one-stranded RNA molecules. Both arms of the pre-miRNA hairpin, denoted as miR-155-5p and miR-155-3p, can give rise to mature miRNAs possessing biological activity.

**Figure 2 ncrna-07-00039-f002:**
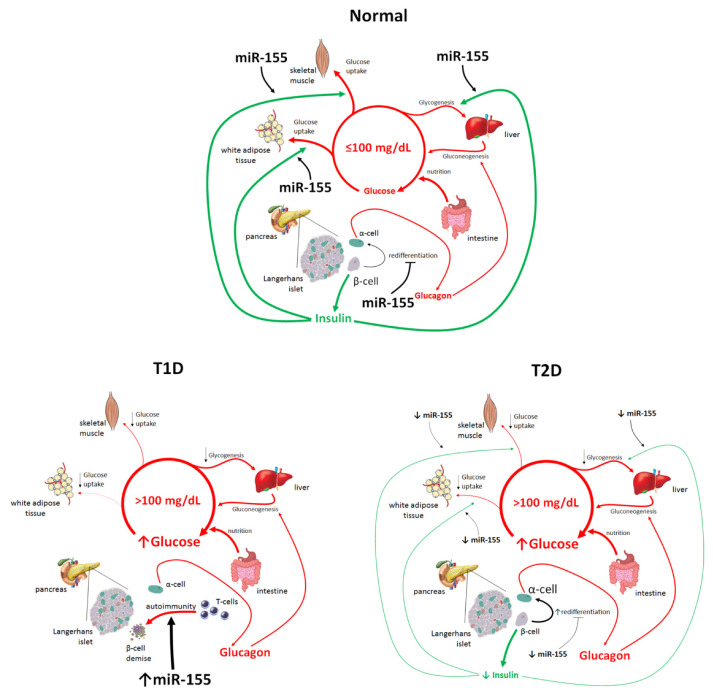
**Functional roles of miR-155 in glucose homeostasis.** The cartoon emphasizes the involvement of miR-155 in the key mechanisms underlying glucose homeostasis: in normal conditions, in type 1 diabetes (T1D) and in type 2 diabetes (T2D); miR-155 improves insulin sensitivity and glucose uptake in white adipose tissue and skeletal muscle and promotes gluconeogenesis in the liver. Moreover, miR-155 prevents the trans-differentiation of β-cells into α-cells under metabolic stress.

**Table 1 ncrna-07-00039-t001:** List of clinical studies examining the association between miR-155 dysregulation and T1D or T2D and their complications. The studies investigating circulating levels of miR-155 are positioned at the top of the table, whereas the studies investigating tissue levels of miR-155 are at the bottom of the table. ↑: upregulation, ↓: downregulation.

Groups Compared	N	Tissue(s)	Main Findings	Ref.
**T2D patients with or without diabetic retinopathy and healthy controls**	170	plasma	↑miR-155 in T2D patients with diabetic retinopathy compared to both controls and T2D patients without retinopathy	[44]
**T1D and age-matched** **healthy controls**	59	plasma	↑miR-155 in T1D patients compared to controls	[45]
**T2D patients with or without diabetic nephropathy**	145	serum	↑miR-155 in patients with diabetic nephropathy compared to T2D patients without diabetic nephropathy miR-155 corelated with microalbuminuria	[46]
**Patients with diabetic nephropathy and** **healthy controls**	38	serum	↑miR-155 in patients with diabetic nephropathy compared to healthy controls	[47]
**Patients with chronic hepatitis C with or without T2D,** **patients with T2D alone and healthy controls**	80	serum	↓miR-155 in T2D patients compared to healthy controls ↓miR-155 in patients with chronic hepatitis C virus infection and T2D compared to patients with chronic hepatitis C virus infection alone	[41]
**T2D patients (grouped** **according to the level of** **albuminuria) and age-matched healthy controls**	83	serum	↓miR-155 in T2D patients regardless of albumin excretion compared to healthy controls	[42]
**T2D and age-matched** **healthy controls**	60	serum	↓miR-155 in T2D patients compared to controls	[40]
**T2D patients without diabetic retinopathy, with** **non-proliferative retinopathy or proliferative retinopathy and healthy controls**	80	serum peripheral white blood cells	↓miR-155 in T2D patients compared to healthy controls ↓miR-155 in T2D patients with non-proliferative retinopathy compared to T2D patients without retinopathy ↓miR-155 in T2D patients with proliferative retinopathy compared to T2D patients with non-proliferative retinopathy	[43]
**T2D and age-matched** **healthy controls**	40	peripheral blood mononuclear cells	↓miR-155 in T2D patients compared to controls	[48]
**T2D and age-matched** **healthy controls**	40	peripheral blood mononuclear cells	↓miR-155 in T2D patients compared to controls	[49]
**T1D patients and** **healthy controls**	959	peripheral white blood cells	miR-155 rs767649 polymorphisms associated with protection from T1D	[50]
**T1D and age-matched** **healthy controls**	86	peripheral blood mononuclear cells	↑miR-155 in T1D patients compared to controls	[51]
**T1D and age-matched** **healthy controls**	41	peripheral blood mononuclear cells	↑miR-155 in T1D patients compared to controls	[52]
**T2D patients with or without diabetic neuropathy,** **healthy controls**	64	peripheral blood mononuclear cells	↓miR-155 in T2D patients with diabetic neuropathy compared to both healthy controls and T2D patients without diabetic neuropathy	[39]
**T2D patients without diabetic retinopathy, with** **non-proliferative retinopathy or proliferative retinopathy and healthy controls**	80	peripheral blood mononuclear cells	↑miR-155 in T2D patients compared to healthy controls ↑miR-155 in T2D patients with non-proliferative retinopathy compared to T2D patients without retinopathy ↑miR-155 in T2D patients with proliferative retinopathy compared to T2D patients with non-proliferative retinopathy	[53]
**Pregnant women with** **gestational diabetes and healthy pregnant women**	69	peripheral white blood cells	↓miR-155 in pregnant women with gestational diabetes compared to healthy controls	[54]
**T2D and age-matched** **healthy controls**	44	platelets	↓miR-155 in T2D patients compared to controls	[55]
**T1D patients with or without diabetic retinopathy and healthy controls**	21	extracellular vesicles from blood	↓miR-155 in T1D patients with diabetic retinopathy compared to both healthy controls and T1D patients without diabetic retinopathy	[56]
**T2D patients and age-matched healthy controls**	15	adipose tissue	miR-155 inversely correlated with mean adipocyte volume and macrophage infiltration	[57]
**Obese and non-obese patients**	50	adipose tissue	↑miR-155 in obese individuals	[58]
**Patients with diabetic nephropathy and** **healthy controls**	9	kidney	↑miR-155 in patients with diabetic nephropathy compared to healthy controls	[59]
**Patients with diabetic nephropathy and** **healthy controls**	98	kidney	↑miR-155 in patients with diabetic nephropathy compared to healthy controls	[60]
**T1D patients with or without diabetic nephropathy and healthy controls**	192	urine kidney	↑miR-155 in patients with diabetic nephropathy compared to healthy controls and T1D patients without diabetic nephropathy	[61]
**T1D patients with or without albuminuria and** **healthy controls**	34	urinary exosomes	↓miR-155 in T1D patients with albuminuria compared both to T1D patients without albuminuria and healthy control	[62]

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
