# Peer review of "Functional Role of miR-155 in the Pathogenesis of Diabetes Mellitus and Its Complications"

_ncrna, 2021, doi:10.3390/ncrna7030039_

Round 1

Reviewer 1 Report

The article maps a new route by emphasizing miR-155 in DM complications. The review is well structured and elegantly presented and the authors are applauded for the well constructed Table and thoughtful figures. Anyone researching ncRNAs and DM knows the importance of miR-155 after decades of studying regulatory mechanisms and DM pathogenesis. The article is likely to be a reference and framework for diabetic complications.

Major point:

The article does not deal with targeting miR-155 in the different tissues relevant for diabetic renal disease, retinopathy and vascular complications and remain an unmet therapeutic need.

Author Response

We thank this Reviewer for her/his careful evaluation of our manuscript and highly valuable comments. We totally agree with the Reviewer that data on targeting miR-155 in tissues relevant for diabetic complications is of great interest. In the revised manuscript, we describe in vivo experiments in which the effects of miR-155 agonists or antagonists were used in order to target diabetic complications; unfortunately, to our knowledge, no human trials were held using miR-155 agonist or inhibitors.

Reviewer 2 Report

the study is interesting and brings an extensive bibliographical review. However, the article does not provide guidance to help the reader on the benefits and harms of miR-155. Different stimuli (glucose or inflammatory cytokines direct different miR-115 responses. In T2D it increases insulin sensitivity, beta cell expansion. Protects from neuropathy but worsens diabetic heart disease. Undefined effects on retinopathy. On the other hand, in T1D it causes apoptosis of beta cells although it favors the differentiation of Tregs. It would be interesting to formulate hypotheses that unite these pathways or justify such different responses in T1D and T2D

Author Response

ANSWER: We truly appreciate the time spent by this Reviewer on our manuscript and we thank her/him for the valuable comments and suggestions. We totally share the Reviewer’s concern about the pleotropic and often very different effects of miR-155 in different tissues, which we better discuss in the revised version of our manuscript. The net effect of miR-155 inhibition or overexpression could be indeed dissimilar in different organs. However, the experimental data from human studies provides us with a clear pattern showing that miR-155 tends to be decreased in patients with T2D and increased in T1D. Such a discrepancy could be related to the different pathogenesis of these two diseases. The findings in animal models are in accordance with clinical trials and may provide the basis for potential mechanistic explanations. We speculate that miR-155 downregulation contributes to the development of insulin resistance in adipose tissue and skeletal muscle and compromise pancreatic insulin secretion by dedifferentiating of beta-cells. Furthermore, mounting evidence indicates that high levels of miR-155 favor progression of a number of autoimmune pathologies, including insulitis. Thus, the therapeutic targeting of miR-155 would pursue different goals in two types of diabetes; in T2D we may use miR-155 agonists, whereas in T1D we may try to use miR-155 inhibitors to eliminate deleterious effects of miR-155 upregulation.

Reviewer 3 Report

The manuscript by Jankauskas and colleagues clearly describes and shows all the evidence which relates miR-155 expression and function to the pathogenesis of Diabetes Mellitus and its complications. The manuscript is well written and structured, and very informative about the multiple roles played by miR-155 in both type 1 and type 2 diabetes.

  • The main change which should be done for this manuscript is to distinguish the studies in which it is reported the differential expression of intra-tissue/cells miR-155 (adipose tissue, PBMCs, platelets, etc.) from the studies in which authors describe the deregulation of circulating miR-155 (plasma, serum, etc.). Indeed, this is an essential difference: differential expression of intracellular miRNAs mirror which is their involvement in diseases pathogenesis; on the other side, deregulated circulating miRNAs are usually described as potential biomarkers for diagnosis, prognosis and monitoring of disease. Therefore, I kindly ask to authors to slightly modify the main text structure and the table 1 by splitting these two different types of studies, and by implementing and deepening the part of biomarker circulating miR-155.
  • Some little language error and misprint should be corrected.
  • In figure 2 it is graphically represented the different role of miR-155 between physiologic conditions and Type 2 Diabetes; however, a lot of information about miR-155 function has also been added for type 1 diabetes. I strongly suggest to implement Figure 2 by graphically adding the role of miR-155 in Type 1 Diabetes.
  • Figures 3 and 4 can be improved. Panels A of both figures are not informative and panels B are too much schematic.

Author Response

ANSWER: We totally agree with this Reviewer’s suggestion and therefore we now clearly distinguish in Table 1 the studies investigating the circulating levels of miR-155 (top of the table) and the studies investigating levels of miR-155 in tissues (bottom of the table). The potential roles of miR-155 as biomarker are now mentioned in the conclusions.

We thank this Reviewer for having carefully read our manuscript. Errors and misprints have been corrected.

We totally agree with this Reviewer; we modified Figure 2 adding the information on T1D.

We agree with the Reviewers that Figures 3 and 4 were not informative (and redundant with the Table) and therefore we decided to remove them.

Round 2

Reviewer 1 Report

Authors have addressed the original comments.